Comparative dosimetric evaluation of single-beam dual-arc versus dual-beam single-arc volumetric modulated arc therapy strategies in lung stereotactic body radiotherapy using Monaco

Ding Jieni 1
Huang Yanqiu 2
Li Qiang 1
Chen Weijun 1
Shao Kainan shaokainan@hmc.edu.cn 1
1 Cancer Center, Department of Radiation Oncology, Zhejiang Provincial People’s Hospital (Affiliated People’s Hospital) , HangZhou , Zhejiang , China
2 Zaozhuang Vocational College , Zaozhuang , Shandong , China
Anson Lesley
Electronic publication date: 2025 Nov 3
Publication date: 2025
Volume: 13
Electronic Location ID: e20311
Received 2025 Apr 22; Accepted 2025 Oct 8
Copyright: ©2025 Ding et al.
Copyright year: 2025
Copyright holder: Ding et al.
License: This is an open access article distributed under the terms of the Creative Commons Attribution License, which permits unrestricted use, distribution, reproduction and adaptation in any medium and for any purpose provided that it is properly attributed. For attribution, the original author(s), title, publication source (PeerJ) and either DOI or URL of the article must be cited.
License URL: https://creativecommons.org/licenses/by/4.0/

Keywords: Monaco, Treatment planning, Optimization, Stereotactic body radiotherapy, Volumetric modulated arc therapy

Funding: Zhejiang Provincial Department of Education General Scientific Research Project No. Y202457103 This work was supported by the Zhejiang Provincial Department of Education General Scientific Research Project (No. Y202457103). The funders had no role in study design, data collection and analysis, decision to publish, or preparation of the manuscript.

==============================
Background and Objective

In stereotactic body radiotherapy (SBRT) for lung cancer, the choice of volumetric modulated arc therapy (VMAT) optimization strategy is critical for achieving optimal target dose coverage while minimizing exposure to normal tissues. This study aims to compare the dosimetric performance and plan complexity of two VMAT optimization strategies in Monaco: single-beam dual-arc (1B2A) versus dual-beam single-arc (2B1A).

Methods

A retrospective analysis was conducted on 50 lung cancer patients treated with SBRT (prescription dose: 50 Gy in 5 fractions). Two VMAT plans were re-optimized using the Monaco treatment planning system: the 1B2A plan (single-beam dual-arc, collimator angle 10°) and the 2B1A plan (dual-beam single-arc, collimator angles 10° and 350°). Dosimetric parameters, including target dose coverage, conformity index (CI), and gradient index (GI), were evaluated for the internal target volume (ITV) and planning target volume (PTV). Dose metrics for organs at risk (OARs) were also analyzed. Plan complexity was assessed based on monitor units (MU), number of control points, complexity index, and integral dose to normal tissues.

Results

Significant dosimetric differences were observed between the two strategies. When normalized to ensure the prescribed 50 Gy isodose line covers 95% of the PTV volume, the high-dose parameters (D1%, D50%, Dmean) of the ITV and PTV were significantly lower in the 1B2A group compared to the 2B1A group (p < 0.001), indicating superior dose distribution with the 2B1A approach. Although the 1B2A plan exhibited marginally better CI, GI, and low-dose lung sparing (V5–V30), these differences were minimal and clinically insignicant. No substantial dierences were found in the dose sparing of other OARs, including the spinal cord, heart, and ribs. Additionally, the 1B2A plan required signicantly higher MU (+15.5%, p < 0.001) and had greater plan complexity (+9.47%, p < 0.001), suggesting lower treatment efficiency.

Conclusions

In peripheral lung cancer SBRT, the dual-beam single-arc (2B1A) strategy offers superior target dose distribution and treatment efficiency, making it a preferable optimization approach.

Introduction

Lung cancer is among the most common malignancies worldwide and remains the leading cause of cancer-related mortality, with non-small cell lung cancer (NSCLC) making up the majority of lung cancer cases (Siegel et al., 2025). For patients with early-stage NSCLC, surgical resection is generally the standard treatment. However, for those who are ineligible for surgery or who decline surgical intervention, stereotactic body radiotherapy (SBRT) has become the preferred nonsurgical alternative (Riely et al., 2024). Characterized by high-dose delivery, a limited number of fractions, and high precision, SBRT increases local control rates while minimizing damage to surrounding normal tissues. Numerous clinical studies have demonstrated that SBRT significantly improves local control in early-stage lung cancer with an acceptable toxicity, and as a result, it is included as a recommended treatment modality in international guidelines (Chang et al., 2015).

As a key technique in SBRT, volumetric modulated arc therapy (VMAT) has rapidly evolved in recent years (Verbakel et al., 2009; Holt et al., 2011). VMAT, an advanced form of intensity-modulated radiation therapy (IMRT), achieves superior dose conformity and treatment efficiency when the gantry speed, dose rate, and multileaf collimator (MLC) position are dynamically adjusted throughout beam rotation (Teoh et al., 2011). Compared with conventional static IMRT, VMAT offers advantages such as improved dose homogeneity and reduced treatment time, effectively mitigating uncertainties caused by respiratory motion and improving clinical outcomes (Ma et al., 2025). The success of SBRT relies heavily on precise dose delivery and effective normal tissue sparing, which VMAT facilitates. With continuous advancements in treatment planning systems (TPSs), VMAT optimization strategies have become increasingly diverse. The choice of optimization strategy critically affects target dose coverage, normal tissue sparing, and overall treatment efficacy (Jiang et al., 2011; Gala et al., 2017; Yang et al., 2020).

The Monaco (Elekta Ltd., Stockholm, Sweden) treatment planning system (TPS) is among the most widely used radiotherapy planning optimization platform in clinical practice. It features a robust Monte Carlo-based dose calculation algorithm and highly flexible optimization parameters, making it extensively utilized in lung cancer SBRT planning (Li et al., 2012; Alhamada et al., 2021). Notably, Monaco’s unique “Number of Arcs” parameter enables the incorporation of two consecutive arcs within a single plane rather than two independent beams. This mechanism divides fluence optimization along the central X-axis, with the first arc optimizing the dose distribution for one half of the target volume and the second arc optimizing the remaining half (Vanetti et al., 2009; Kalet et al., 2017; Clements et al., 2018). Theoretically, this approach increases dose modulation precision. However, its dosimetric performance and plan complexity in lung cancer SBRT remain insufficiently studied. Moreover, quantitative comparisons of dosimetric and complexity differences between single-beam multi-arc (e.g., 1B2A) and multi-beam single-arc (e.g., 2B1A) optimization strategies are limited. As a result, clinical physicists often rely on personal experience rather than rigorous scientific evidence when selecting VMAT optimization strategies, thereby introducing variability in treatment plan quality that may ultimately impact clinical outcomes. Therefore, a systematic investigation is needed to clarify the differences in target dose coverage, normal tissue sparing, and plan complexity between different optimization strategies, providing clinical physicists with a solid evidence-based foundation for decision-making.

This study focuses on two VMAT optimization strategies in the Monaco TPS: single-beam dual-arc (1B2A) and dual-beam single-arc (2B1A) strategies. A retrospective analysis was conducted to compare the dosimetric characteristics and plan complexity of lung cancer SBRT treatment plans. The objective is to systematically evaluate and compare their performance across multiple dimensions, including target dose coverage, normal tissue sparing, and plan complexity, to delineate the clinical advantages and limitations of each strategy and provide evidence-based guidance for clinical treatment planning. Through comprehensive dosimetric evaluation and complexity analysis, the aim of this study is to assist clinicians in optimizing the balance between target coverage, organ-at-risk (OAR) protection, and treatment efficiency to ultimately improve the quality of lung cancer SBRT plans.

Materials and Methods

A total of 50 patients were included in this study; the patients had peripheral lung cancer and underwent SBRT at our institution between January and December 2023, with a prescribed dose of 50 Gy in five fractions. Patients were excluded if they had not received radiotherapy at our institution or if their prescribed dose did not meet the study criteria. Because this was a retrospective study, all patient data were anonymized, and approval was obtained from the Medical Ethics Committee of Zhejiang Provincial People’s Hospital (2024090). The study was conducted in accordance with the ethical standards of the Declaration of Helsinki. Patient consent was waived by the Medical Ethics Committee of Zhejiang Provincial People’s Hospital because of the anonymity of the data. All patient computed tomography (CT) images were anonymized at the time of data collection. The subsequent comparisons in this study were conducted using newly reoptimized radiotherapy plans on Monaco, not the original plans utilized in clinical treatment.

During prior radiotherapy procedures, all enrolled patients underwent a simulation using a large-bore CT simulator (GE Healthcare) in the supine position, which was immobilized with an integrated board and thermoplastic mask, with both arms raised above the head. The scan range extended from the superior border of the second cervical vertebra (C2) to the inferior border of the second lumbar vertebra (L2), with a slice thickness of three mm. Four-dimensional (4D) CT scanning was performed, and the acquired CT images were imported into the Monaco treatment planning system in the DICOM format. To define the internal target volume (ITV), 10 respiratory phases (ranging from 0% to 90%) were used in the 4D-CT scan. The target volumes and organs at risk (OARs), including the lungs, heart, spinal cord, and ribs, were delineated by experienced radiation oncologists. The ITV was generated using a 4D maximum intensity projection (MIP) (Keall et al., 2006), and the planning target volume (PTV) was created by adding a symmetric five mm margin around the ITV, in accordance with RTOG-0813 guidelines (Bezjak et al., 2019). This retrospective study analyzed only imaging data and target delineations, with both treatment plans retrospectively reoptimized.

The radiotherapy plans in this study were optimized using the Monaco (version 6.0) treatment planning system on the Infinity LINAC (Elekta Ltd., Stockholm, Sweden) with Agility MLC and 6 MV X-rays in flattening filter-free (FFF) mode (Gholampourkashi et al., 2019). For the left lung, the beam (named ‘A1’) angle was set to CCW 178°, with an arc length of 216° and a collimator angle of 10°. For the right lung, the beam angle was set to CCW 38°, with an arc length of 216° and a collimator angle of 10°. A tight (two mm) margin was applied to the target, with an increment (Inc) set to 12, a maximum of 240 control points per arc, two arcs, a calculation grid resolution of 2.5 mm, and a Monte Carlo calculation uncertainty of 0.5% per calculation.

During plan design and optimization, a standardized optimization approach was applied. For each patient, first, the 1B2A plan was created using the initial conditions shown in Fig. 1. The plan underwent multiple rounds of Phase 1 fluence optimization to determine the most suitable dose constraints on the basis of the patient’s anatomy and clinical needs. These optimized dose constraints were then considered, and the most appropriate plan for each patient was selected. Once the optimization conditions were finalized for the 1B2A plan through Phase 1 fluence optimization, the same dose constraints were applied to the 2B1A plan. The 2B1A plan was created by duplicating the 1B2A plan and modifying only the arc configuration (i.e., the number of arcs modified to one, the ‘A1’ beam was copied and set to the CW direction, with the collimator angle adjusted to 350° as beam ‘A2’). Both plans were then subjected to the following steps: first, the calculation engine was first reset, which was followed by one round of Phase 1 fluence optimization, one round of segment optimization, and final normalization, ensuring that 95% of the PTV was covered by the prescribed 50 Gy dose. This process ensured that both plans were optimized under identical conditions, with the only difference being the arc configuration.

Figure 1 Example of individualized IMRT constraints for the 1B2A plan in the Monaco TPS, detailing the specific dose constraints applied to the target (PTV) and normal tissues (body, ribs, and lungs), including the constraint type, weight, reference dose, power law exponent, and shrink margin settings.

The radiotherapy plans were evaluated on the basis of the Timmann criteria (Timmerman, 2022) and the RTOG-0813 protocol. The target volume assessments included D2%, D98%, D50%, and Dmean for the clinical target volume (CTV) and planning target volume (PTV), as well as the conformity index (CI) and gradient index (GI). The CI was calculated using the Paddick formula (Paddick, 2000) CI = (TVPIV)2/(TV∗PIV), where TVPIV is the volume of the target encompassed by the prescription isodose line, TV is the target volume, and PIV is the total volume receiving the prescription dose. The GI was defined as the ratio of the 50% prescription isodose volume (25 Gy) to the 100% prescription isodose volume (50 Gy), representing the dose fall-off beyond the target (Paddick & Lippitz, 2006). Additionally, the maximum dose at two cm beyond the PTV (D2 cm) was included in the evaluation. To assess the dose received by normal tissues, the normal tissue integral dose (NTID) was calculated as the mean dose multiplied by the volume of normal tissue outside the PTV, following the method described by Aoyama et al. (2006). The plan quality was assessed using monitor units (MUs) and the number of control points (CP). Additionally, plan complexity was quantified using the edge metric for a comprehensive evaluation of treatment plan optimization (Younge et al., 2016). The edge metric complexity was calculated as follows: M=1MU∑i=1NMUi×yiAi

where MU represents the total number of monitor units in the plan, MUi is the number of monitor units delivered through aperture i, Ai is the open area of aperture i, and yi is the aperture perimeter excluding the MLC leaf ends.

Statistical analyses were conducted using SPSS 26.0. Continuous variables were reported as the mean ± standard deviation (mean ± SD), and intergroup comparisons were performed using paired t tests or Wilcoxon signed-rank tests, depending on data normality. A significance threshold of P < 0.05 was applied. Non-statistically analyzed dose metrics are provided in the Supplemental Information. See Supplemental Information for full OAR dose metrics.

Results

The patients’ tumors were classified by location into the left or right lung and divided into upper and lower lobes. The tumor distribution was as follows: 20% (20 patients) had tumors in the left upper lobe, 6% (six patients) in the left lower lobe, 10% (10 patients) in the right upper lobe, 8% (eight patients) in the right middle lobe, and 6% (six patients) in the right lower lobe. The mean ITV was 7.42 ± 4.80 cm3, and the mean PTV was 22.03 ± 11.38 cm3.

In this study, the dosimetric outcomes of 50 lung cancer patients treated with stereotactic body radiotherapy (SBRT) were retrospectively analyzed, and two VMAT optimization strategies, single-beam dual-arc (1B2A) and dual-beam single-arc (2B1A), were compared. The 1B2A strategy utilized a single beam with a collimator angle of 10° and a dual-arc configuration (number of arcs = 2), whereas the 2B1A strategy employed two beams with collimator angles of 10° and 350° using a single-arc configuration (number of arcs = 1). Both plans were optimized using the Monaco system under identical settings, differing only in their arc configuration and beam number. All VMAT arcs used a bidirectional arc technique, with arc angles tailored to the PTV location. After all the plans were normalized to ensure that 95% of the PTV was covered by the prescription dose of 50 Gy, the following differences were observed between the single-beam dual-arc (1B2A) strategy and the dual-beam single-arc (2B1A) strategy:

• ITV dose parameters (D99%, D95%, D50%, Dmean, D1%) were lower in the 1B2A group than in the 2B1A group.

• No significant differences were found in low-dose target coverage (PTV_D99%) between the two groups. However, the mid-to-high dose parameters (D50%, Dmean, D1%) for PTV were significantly lower in the 1B2A group, and PTV_D0.03cc was reduced by 0.75% in the 1B2A group (p < .001), suggesting lower high-dose deposition in the 1B2A plan.

The dosimetric differences between the 1B2A and 2B1A plans in terms of the ITV and PTV parameters are summarized in Table 1 and Figs. 2 and 3. The average dose volume histograms of the ITV and PTV for 50 patients are shown in Fig. 4. The left-hand figure presents the complete dose range (0 ∼ 70 Gy), while the right-hand figure magnifies the 50 ∼ 70 Gy dose interval, highlighting the dosimetric differences in the high-dose region.

Table 1 Dosimetric comparison of ITV and PTV parameters between single-beam dual-arc (1B2A) and dual-beam single-arc (2B1A) VMAT plans in lung SBRT. (Doses are expressed as mean ± SD in cGy.).

Evaluations	P Value	1B2A	2B1A	Difference	
ITV D99%(cGy)	.006	5,557.4 ± 266.86	5,604.6 ± 204.5	−47.2	
ITV D95%(cGy)	<.001	5,671.8 ± 262.24	5,731.8 ± 194.31	−60	
ITV D50%(cGy)	<.001	6,060.9 ± 240.93	6,141.5 ± 152.02	−80.6	
ITV Dmean(cGy)	<.001	6,058.0 ± 233.22	6,134.2 ± 149.26	−76.2	
ITV D1%(cGy)	<.001	6,532.7 ± 209.1	6,589.9 ± 148.51	−57.2	
PTV D99%(cGy)	n.s.	4,844.2 ± 27.656	4,843.1 ± 26.187	1.1	
PTV D50%(cGy)	.010	5,627.9 ± 88.311	5,652.5 ± 60.293	−24.6	
PTV Dmean(cGy)	.0011	5,638.9 ± 88.828	5,669.5 ± 53.024	−30.6	
PTV D1%(cGy)	<.001	6,471.8 ± 202.06	6,537.5 ± 139.9	−65.7	
PTV D0.03cc(cGy)	<.001	6,564.9 ± 187.18	6,614.8 ± 133.37	−49.9	
PTV CI50Gy	.0025	0.87325 ± 0.024472	0.86961 ± 0.025385	0.00364	
PTV GI25Gy	.018	4.8426 ± 0.61054	4.8693 ± 0.62865	−0.0267	
D2cm D0.03cc(cGy)	n.s.	3,355.0 ± 1,890.4	3,365.8 ± 1,905.8	−10.8	

Figure 2 Comparison of the ITV dosimetric parameters between the two VMAT optimization strategies (1B2A and 2B1A) in lung cancer SBRT.

The box plots illustrate the ITV dose values for D99%, D95%, D50%, Dmean, and D1% (unit: cGy), with the statistical significance levels indicated.

Figure 3 Comparison of PTV dosimetric parameters between the two VMAT optimization strategies (1B2A and 2B1A) in lung cancer SBRT.

The figure presents the PTV dose values for D99%, D50%, Dmean, D1%, and PTV_D0.03cc (unit: cGy). The notation “n.s.” indicates no statistically significant difference between the groups, while all other marked parameters show statistically significant differences.

Figure 4 Average dose-volume histogram (DVH) comparison of target volumes (ITV and PTV) between the two VMAT optimization strategies (1B2A and 2B1A) for 50 patients.

The left panel displays the full dose range (0–70 Gy), while the right panel provides a magnified view of the 50–70 Gy dose interval, highlighting the dosimetric differences in the high-dose region.

The PTV_CI50Gy, which reflects dose conformity within the target, was slightly higher in the 1B2A group (0.873) than in the 2B1A group (0.870), indicating marginally improved high-dose conformity in the 1B2A plan, although the difference was minimal. The PTV_GI25Gy, which quantifies dose fall-off from the target to normal tissues, was slightly lower in the 1B2A group than in the 2B1A group (4.843 vs. 4.869). The absolute difference (0.0267) is minimal and falls within the range of expected variation that has not been shown to significantly impact clinical outcomes in prior studies. Additionally, both optimization strategies exhibited similar control over D2 cm_D0.03cc, the maximum dose at two cm beyond the target, indicating negligible differences in dose attenuation at greater distances from the target.

The Lungs V0Gy, V30Gy, V10Gy, and V5Gy values in the 1B2A group were slightly lower than those in the 2B1A group. Although some differences were statistically significant (p < .05), the absolute differences were minimal. For example, the difference in lung V20Gy between groups was 0.0341%, far below the 10% clinical threshold for pneumonitis risk, indicating limited clinical impact despite statistical significance. Lungs_Dmean did not significantly differ, indicating that both plans achieved similar overall lung mean dose control. The maximum doses (D0.03cc) for the heart, spinal cord, and chest wall were comparable between the two groups, suggesting that both optimization strategies provided equivalent dose control for these critical organs. A detailed comparison of OAR dose-volume parameters for both VMAT strategies is presented in Table 2 and Figs. 5 and 6.

Table 2 Dosimetric comparison of OAR parameters between single-beam dual-arc (1B2A) and dual-beam single-arc (2B1A) VMAT plans in lung SBRT.

Evaluations	P Value	1B2A	2B1A	Difference	
Lungs V5Gy(%)	n.s.	14.014 ± 4.6249	14.036 ± 4.6381	−0.022	
Lungs V10Gy(%)	n.s.	8.6795 ± 3.6918	8.6524 ± 3.6758	0.0271	
Lungs V20Gy(%)	<.001	3.8401 ± 1.9241	3.8742 ± 1.962	−0.0341	
Lungs V30Gy(%)	<.001	1.9897 ± 1.0693	2.0108 ± 1.0899	−0.0211	
Lungs Dmean(cGy)	n.s.	314.33 ± 106.25	315.15 ± 107.32	−0.82	
Heart D0.03cc(cGy)	n.s.	1,308.3 ± 1,450.1	1,298.7 ± 1,453.6	9.6	
SpinalCord D0.03cc(cGy)	n.s.	1,099.6 ± 550.22	1,075.6 ± 568.49	24	
Ribs D0.03cc(cGy)	n.s.	4,005.1 ± 1,209.4	3,999.9 ± 1,209.7	5.2	

Figure 5 Comparison of lung dose-volume parameters between the two VMAT optimization strategies (1B2A and 2B1A).

The box plots display Lungs V5Gy, V10Gy, V20Gy, and V30Gy (unit: %). The notation “n.s.” indicates no statistically significant difference between groups, whereas the other parameters are significantly different (p < .05).

Figure 6 Comparison of dosimetric parameters for major OARs (lungs, heart, spinal cord, and ribs) between the two VMAT optimization strategies (1B2A and 2B1A).

The figure presents Lungs Dmean, Heart D0.03 cc, SpinalCord D0.03 cc, and Ribs D0.03 cc (unit: cGy). None of these parameters showed statistically significant differences (“n.s.”).

The MU value in the 1B2A group was significantly greater than that in the 2B1A group (+15.5%, p < .001), suggesting that the 1B2A plan required greater dose delivery, potentially leading to longer treatment times. The number of control points (CPs) in the 1B2A group was 7.23% higher than that in the 2B1A group (p < .001), indicating increased plan complexity. This suggests that the 1B2A plan involved more intricate MLC movements, potentially prolonging treatment time. The normal tissue integral dose (RVR NTID) was also slightly higher in the 1B2A group than in the 2B1A group, indicating differences in the overall normal tissue dose burden between the two optimization strategies. This may be associated with the higher MU consumption observed in the 1B2A group. Plan complexity parameters, including monitor units (MUs), complexity indices, and control points, significantly differed between the 1B2A and 2B1A strategies, as shown in Table 3.

Table 3 Comparison of plan complexity parameters between single-beam dual-arc (1B2A) and dual-beam single-arc (2B1A) VMAT plans in lung SBRT.

Evaluations	P Value	1B2A	2B1A	Difference	
plan mus	<.001	2,915.7 ± 523.62	2,525.4 ± 391.86	390.3	
complexity (mm−1)	<.001	0.11276 ± 0.018062	0.10126 ± 0.016498	0.0115	
plan cps	<.001	210.72 ± 14.799	196.52 ± 15.273	 14.2	
RVR NTID (Gy*cm3)	.019	2415,800.0 ± 712,530.0	2426,300.0 ± 717,040.0	−1,0500	

Discussion

In this study, 50 lung cancer patients treated with SBRT retrospectively analyzed, and the dosimetric characteristics and plan complexity of two VMAT optimization strategies in the Monaco treatment planning system (TPS), namely, single-beam dual-arc (1B2A) and dual-beam single-arc (2B1A), were compared. Under the same PTV coverage conditions (50 Gy in 5 fractions, with 95% of the PTV receiving the prescribed dose), the 1B2A strategy resulted in lower dose deposition in high-dose regions but did not offer significant advantages in target conformity or normal tissue sparing. Furthermore, it increased plan complexity and monitor unit (MU) consumption, potentially extending the treatment time. These findings suggest that the clinical benefit of 1B2A is limited, particularly for simple targets, where the 2B1A approach may be more clinically feasible.

In SBRT dosimetric planning, high-dose coverage is critical for improving tumor control. International clinical guidelines, including RTOG 0813, recommend increasing local high-dose regions within safe limits to increase tumor control. Furthermore, the use of controlled high-dose hotspots (e.g., restricting the maximum PTV dose to 140% of the prescribed dose, or 70 Gy in this study) can improve treatment efficacy. Given its superior high-dose coverage, the 2B1A strategy may be preferable for lung cancer SBRT patients who require improved target coverage and dose escalation.

Although the 1B2A strategy slightly reduced lung exposure in the low-dose region (V20Gy, V30Gy), the absolute differences were minimal (e.g., V20Gy: 0.0341%) and were well below the clinically established risk thresholds (e.g., V20Gy ¡10%), indicating limited clinical impact. Additionally, the dose to the spinal cord (D0.03cc) was slightly greater in the 1B2A group, which further reduced its potential advantage in terms of normal tissue sparing. The increase in MU values and plan complexity suggests a potential prolongation of treatment time and an elevated risk of delivery errors, which may negatively affect treatment quality in clinical practice.

In this study, plan complexity was comprehensively evaluated using MU values, the number of control points (CP), and the complexity index. Higher MU values indicate longer dose delivery times, which may increase patient discomfort and the risk of setup errors. Additionally, higher plan complexity can make dose verification and quality assurance (QA) more challenging. Previous research has shown a strong correlation between plan complexity, treatment error risk, QA difficulty, and plan execution stability. Thus, when an optimization strategy is selected, the balance between complexity and treatment quality must be carefully considered (Miften et al., 2018; Quintero et al., 2021; Radici et al., 2024).

The dosimetric and complexity results in this study differ from those of several previous studies. For example, Kalet et al. (2017) analyzed 17 pelvic cases and reported that for complex-shaped targets, single-beam multi-arc optimization provided better dosimetric performance and conformity advantages while also reducing the MU values and the number of control points. Their study emphasized that when multiple arcs (e.g., two arcs) were used per treatment session, the Monaco system divided fluence optimization along the central X-axis, optimizing half of the target volume per arc rotation. This approach might be advantageous for complex target geometries. However, in this study’s lung cancer SBRT setting with simpler targets, the single-beam dual-arc strategy showed limited clinical applicability.

This discrepancy may be closely related to tumor location and target complexity. In this study, all tumors were peripheral lung lesions, which are typically small, well defined, and relatively isolated. Under these conditions, the dual-beam single-arc (2B1A) strategy was sufficient to achieve optimal dose distribution and conformity. However, as indicated by previous studies (e.g., Kalet et al., 2017), for more complex tumor geometries—such as centrally located tumors or lesions adjacent to critical structures—the single-beam dual-arc (1B2A) strategy may offer advantages in improving target conformity and dosimetric precision. Therefore, in future clinical applications, the selection of a VMAT optimization strategy should be tailored to the geometric complexity of the target volume. Further studies focusing on complex tumor sites are warranted to clarify the potential benefits of 1B2A in such scenarios.

In this study, actual beam-on time or delivery accuracy were not evaluated, both of which are important clinical considerations. Although MUs and control points serve as indirect indicators, the actual delivery time can vary depending on the treatment system efficiency, collimator rotation speed, and patient setup. Higher MUs and plan complexity, as observed with the 1B2A strategy, may prolong treatment duration, potentially affecting patient comfort and increasing susceptibility to motion-related uncertainties. Additionally, higher complexity may pose challenges for QA and delivery accuracy, which were not assessed in this retrospective dosimetric comparison. These aspects should be explored in future prospective studies to assess the full clinical impact of different VMAT strategies. For the 2B1A configuration, composite field sequencing can be used when the plan is transmitted to Mosaiq, which generates a single beam for treatment, minimizing the time difference in treatment direction changes compared to the 1B2A approach. As highlighted by Panizza et al. (2023), this additional setup time can range from 30 to 40 s, depending on the treatment system used.

The limitations of this study also warrant further discussion. First, the sample size was relatively small (50 cases), and the retrospective design limits the ability to assess the long-term effects of optimization strategies on clinical outcomes, such as local control rates, survival, and toxicity. Additionally, this study was conducted at a single center, which may limit the generalizability of the findings to other clinical settings and treatment planning systems (e.g., Eclipse and RayStation). Future studies with larger, multicenter cohorts and prospective designs are needed to confirm these findings and assess their applicability across different TPS systems. Moreover, this study did not account for treatment uncertainties such as respiratory motion, setup errors, or dose delivery accuracies, all of which could influence treatment planning and delivery. These factors are especially important in clinical practice, where real-time adjustments are made to account for patient movement and anatomical changes. The retrospective nature of the study means that the plans were not adjusted for real-time patient motion or positioning uncertainties, which are commonly managed in clinical practice through techniques such as 4DCT and respiratory gating. Additionally, treatment delivery in real time may differ from what was planned because of various clinical factors, including anatomical changes, patient movement, and machine-specific factors. Notably, while this study provides valuable dosimetric insights, the clinical utility of the findings should be interpreted with caution, as patient outcome data, including tumor control, survival rates, and toxicity, were not included in the analysis. Future prospective studies incorporating clinical outcomes will be necessary to evaluate the true impact of these optimization strategies on patient care. In future studies, larger prospective trials are needed to explore the effects of optimization strategies on clinical efficacy and toxicity outcomes. The incorporation of real-time motion management, dose verification during treatment, and QA assessments would offer a more comprehensive understanding of the clinical utility of VMAT optimization strategies, increasing their applicability in routine clinical practice.

Conclusion

In this study, systematically compared single-beam dual-arc (1B2A) and dual-beam single-arc (2B1A) VMAT optimization strategies in lung cancer SBRT using the Monaco TPS. The findings demonstrated that in simple-target settings, the single-beam dual-arc strategy provided limited dosimetric benefits while increasing treatment complexity. On the basis of these results, the dual-beam single-arc strategy should be prioritized in clinical practice for lung cancer SBRT planning. This study provides quantitative, objective evidence to support clinical decision-making, which may contribute to optimizing SBRT planning strategies and improving the clinical efficacy and patient outcomes of lung cancer SBRT.

Supplemental Information

Supplemental Information 1 Raw dosimetric data of 1B2A group collected in this study

Supplemental Information 2 Raw dosimetric data of 2B1A group collected in this study

Supplemental Information 3 STROBE checklist

The authors would like to thank AJE for English language editing services.

Additional Information and Declarations

Competing Interests

Author Contributions

Human Ethics

Data Availability

The authors declare there are no competing interests.

Jieni Ding performed the experiments, authored or reviewed drafts of the article, and approved the final draft.

Yanqiu Huang performed the experiments, prepared figures and/or tables, and approved the final draft.

Qiang Li analyzed the data, prepared figures and/or tables, and approved the final draft.

Weijun Chen conceived and designed the experiments, authored or reviewed drafts of the article, and approved the final draft.

Kainan Shao conceived and designed the experiments, analyzed the data, authored or reviewed drafts of the article, and approved the final draft.

The following information was supplied relating to ethical approvals (i.e., approving body and any reference numbers):

The Medical Ethics Committee of Zhejiang Provincial People’s Hospital.

The following information was supplied regarding data availability:

The raw data collected in this study are available in the Supplementary Files.

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
