# Peer review of "Comparative dosimetric evaluation of single-beam dual-arc versus dual-beam single-arc volumetric modulated arc therapy strategies in lung stereotactic body radiotherapy using Monaco"

_PeerJ, doi:10.7717/peerj.20311_

## Round 0.1 · original submission · Major Revisions

· Academic Editor

Major Revisions

**Language Note:** When you prepare your next revision, please either (i) have a colleague who is proficient in English and familiar with the subject matter review your manuscript, or (ii) contact a professional editing service to review your manuscript. PeerJ can provide language editing services - you can contact us at [email protected] for pricing (be sure to provide your manuscript number and title). – PeerJ Staff

Reviewer 1 ·

Basic reporting

-

Experimental design

-

Validity of the findings

-

Additional comments

This retrospective dosimetric study evaluates two volumetric modulated arc therapy (VMAT) optimization strategies—single-beam dual-arc (1B2A) and dual-beam single-arc (2B1A)—in 50 patients undergoing stereotactic body radiotherapy (SBRT) for peripheral lung cancer using the Monaco treatment planning system (TPS). The primary aim was to compare target coverage, organ-at-risk (OAR) sparing, and plan complexity between the two strategies. The topic is timely and clinically relevant, particularly for centers using the Monaco TPS for SBRT. The manuscript is well-structured, the methodology is appropriately designed and described, and the findings are clearly reported. The results demonstrate that the 2B1A approach offers superior high-dose coverage and greater treatment efficiency with reduced plan complexity, while 1B2A confers only minimal advantages in low-dose sparing. The study fills a specific technical gap in the literature and provides practical planning insights. However, some revisions are required to enhance clarity, contextual interpretation, and conciseness.

Major Issues
• While the data are statistically significant, the authors should better contextualize the clinical relevance of differences (e.g., small variations in V20 or GI). Statements like “clinically insignificant” are used, but without quantitative thresholds or reference to clinical impact.
• The discussion should also emphasize that results may be specific to simple or peripheral lesions, and 1B2A could still be useful in more complex geometries.
• Although MU and control points are discussed, beam-on time, patient comfort, and delivery accuracy are not addressed. Even an estimated discussion would improve clinical applicability.
• Similarly, plan complexity is mentioned, but QA deliverability or pass rates are not evaluated or cited from prior studies.
• The study lacks a detailed description of tumor location, proximity to critical structures, or shape complexity, which could affect optimization strategy selection. Including a summary table or figure of tumor site distribution (e.g., left vs. right lung, upper vs. lower lobe) would aid interpretation.
• The study mentions limitations but could expand briefly on motion management, treatment uncertainties, and how retrospective planning might differ from real-time clinical planning.
• Both the Abstract and Discussion sections contain redundant phrases reiterating the same result (e.g., “superior dose distribution,” “improved efficiency,” “preferable optimization approach”). These should be condensed.

Minor Issues
• Clarify technical terms like “Edge Metric” and “NTID” for general readers unfamiliar with TPS-specific metrics.
• The manuscript is generally well-written, but a light English editing pass is recommended to improve fluency and reduce wordiness.
• The supplementary files are appropriate, but a note in the main text should more clearly reference where they support specific results (e.g., “see Supplemental Table S1 for full OAR dose metrics”).

·

Basic reporting

The authors conducted a retrospective analysis of 50 lung cancer patients treated with SBRT, comparing two different planning techniques (1B2A and 2B1A) across multiple parameters, including target dose coverage, normal tissue sparing, and plan complexity. The topic is of interest to the medical physics community, particularly for Monaco users, given the limited literature on optimal arc geometry. However, the clinical utility of the study should be interpreted with caution, as patient outcome data were not included in the analysis. The structure of the manuscript would also benefit from revision. For instance, lines 141–146 are more appropriate for the Results section, and lines 148–154 appear redundant and should be more concisely summarized. Overall, the Results section is overly descriptive, whereas the tables are more informative and easier to interpret.

Experimental design

Could the authors clarify why a 3 mm slice thickness was chosen for CT acquisition in the SBRT setting? This resolution may be suboptimal for accurate target delineation and dose calculation in lung SBRT.

Please specify how many respiratory phases were used in the 4D-CT for ITV definition.

For improved clarity, I suggest reporting all margins in millimeters (mm) rather than cm.

The phrase “strict control variable approach” is somewhat vague. As a Monaco user, I would point out that even when using the same planning parameters, different optimization runs can yield varying results due to the stochastic nature of the Monte Carlo algorithm. This is an inherent limitation in comparative planning studies and should be acknowledged. How did the authors ensure that the two plans differed solely in arc geometry? Were multiple optimization attempts performed for each plan? If so, how many trials were needed to achieve clinically acceptable results, and were the best plans selected for comparison? More methodological detail would be appreciated.

Validity of the findings

The analysis is overall well presented; however, the authors should acknowledge that the statement regarding extended treatment time with single-beam dual-arc techniques may not be entirely accurate. In clinical practice, the additional time required to register the second beam, rotate the collimator, and initiate delivery needs to be taken into account. For instance, in our department using a VersaHD linac, we estimated this additional time to be approximately 30/40 seconds. As such, we often prefer single-beam single-arc plans to optimize treatment efficiency. This observation is supported by Panizza et al. (https://doi.org/10.3390/cancers16010013), and we suggest the authors include this point as a limitation of the time analysis presented.

Reviewer 3 ·

Basic reporting

The manuscript entitled “Dosimetric Comparison Between Monaco Single-Beam Dual-Arc and Dual-Beam Single-Arc VMAT Optimization in Lung SBRT: A Retrospective Study” has the following comments for corrections before publication in PeerJ.

1. The title is clear and concise, but could be improved for specificity. Suggested title: “Comparative Dosimetric Evaluation of Single-Beam Dual-Arc Versus Dual-Beam Single-Arc VMAT Strategies in Lung SBRT Using the Monaco TPS”

2. The "complexity index" is cited but lacks a precise formula or reference in the main text. Clarify how the complexity metric is defined (Edge Metric?), citing Younge et al. (2016), and provide its formula or calculation method.

3. All plans are normalized to 95% PTV coverage at 50 Gy. However, since 1B2A and 2B1A differ in beam arrangement, this could be a biased interpretation. Include justification for this normalization strategy. Discuss whether the same normalization favors one method in terms of MU or dose spill.

4. The Introduction section is well-structured but overly long. Consider splitting into (Clinical background - VMAT and Monaco TPS technical overview - Study rationale)

5. The methods section has a very good level of detail. However, Figure 1 is referenced but not visible in the submitted version. Ensure all figures are properly embedded.

6. In the results section, some tables repeat in figures (e.g., DVH metrics and box plots). Consider streamlining presentation: retain one detailed table and one summarizing figure per section (ITV/PTV, OARs, complexity). Tables 1-3 should use consistent units (e.g., Gy or cGy).

7. The discussion has excellent comparative insight but lacks mention of plan deliverability or QA feasibility in high-complexity cases, and respiratory motion management, and whether arc types influence motion robustness.

8. Generally, the comparison between 1B2A and 2B1A in the Monaco TPS context is important and clinically relevant. The study is valuable for centers using Monaco TPS, but should emphasize its single-center, retrospective nature more explicitly in limitations, and discuss generalizability to other TPS systems (e.g., Eclipse, RayStation).

Experimental design

-

Validity of the findings

-

---

## Round 0.2 · accepted · Accept

· Academic Editor

Accept

Thank you for the revisions you have made to your manuscript, which all three reviewers have now approved for publication.

Reviewer 1 ·

Basic reporting

Please see below.

Experimental design

Please see below.

Validity of the findings

Please see below.

Additional comments

The authors have satisfactorily addressed all reviewer comments, including clarifying the clinical relevance of small dosimetric differences with quantitative thresholds, expanding the discussion on applicability to complex tumor geometries, and providing additional methodological details (tumor distribution, 4D-CT phases, CT slice thickness, margins, and optimization process). They justified the normalization strategy, added the Edge Metric formula and references, and expanded on plan complexity, QA, and treatment delivery considerations with appropriate citations. Structural improvements were made by reorganizing the Introduction, condensing the Abstract and Discussion, streamlining the Results, and embedding missing figures, while the manuscript also underwent professional English editing. Supplementary files are now clearly referenced. Overall, the revision has improved clarity, rigor, and readability, and I recommend acceptance in its current form.

·

Basic reporting

All the issues raised have been addressed by the authors, and I do not have any major concerns.

Experimental design

All the issues raised have been addressed by the authors, and I do not have any major concerns.

Validity of the findings

All the issues raised have been addressed by the authors, and I do not have any major concerns.

Additional comments

All the issues raised have been addressed by the authors, and I do not have any major concerns.

Reviewer 3 ·

Basic reporting

According to the sufficient reply of authors to the comments of the reviewers, the paper now is ready for publication in PeerJ.

Experimental design

According to the sufficient reply of authors to the comments of the reviewers, the paper now is ready for publication in PeerJ.

Validity of the findings

According to the sufficient reply of authors to the comments of the reviewers, the paper now is ready for publication in PeerJ.